# Removing Acrylic Conformal Coating with Safer Solvents for Re-Manufacturing Electronics

**DOI:** 10.3390/polym13060937

**Published:** 2021-03-18

**Authors:** Taofeng Lu, Gregory Reimonn, Gregory Morose, Evan Yu, Wan-Ting Chen

**Affiliations:** 1Department of Plastics Engineering, Francis College of Engineering, University of Massachusetts Lowell, Lowell, MA 01854, USA; Taofeng_Lu@student.uml.edu (T.L.); Gregory_Reimonn@student.uml.edu (G.R.); Evan_Yu@student.uml.edu (E.Y.); 2Toxics Use Reduction Institute, University of Massachusetts Lowell, Lowell, MA 01852, USA; Gregory_Morose@uml.edu

**Keywords:** solvent, coating stripper, methylene chloride, safety, re-manufacturing, circular economy

## Abstract

Conformal coating is typically composed of polymeric film and is used to protect delicate electronic components such as printed-circuit boards. Without removing conformal coating, it would be difficult to repair these complicated electronics. Methylene chloride, also called dichloromethane (DCM), has a widespread usage in conformal coating stripper products. The high toxicity of DCM increases human health risk when workers are exposed to DCM during the conformal coating removal processes. Therefore, the replacement of DCM would be beneficial to greatly improve the overall safety profile for workers in the electronics and coating industries. This research identified and evaluated alternative chemicals for replacing DCM used in acrylic conformal coating stripping operations. The solubility of an acrylic conformal coating was measured and characterized using Hansen solubility parameters (HSP) theory. Coating dwell time tests using various solvent blends verified the accuracy of the created HSP solubility sphere. A data processing method was also developed to identify and screen potential alternative solvent blends in terms of safety, toxicity, and cost-effectiveness. The identified safer solvent blends were demonstrated to provide equivalent stripping performance as compared to DCM based coating strippers within an acceptable cost range. The results of this research will be of value to other types of conformal coatings, such as silicone and polyurethane, where DCM is commonly used in similar coating stripping operations. By safely removing conformal coating, delicate electronics would be available for re-manufacturing, enabling a circular economy.

## 1. Introduction

Conformal coatings are used to coat and protect printed circuit boards and the electronic components mounted on the boards. As a general production step to improve the durability of printed circuit boards, conformal coatings provide moisture resistance as well as dust and abrasion resistance, contributing to longer product life and reduced maintenance cost. Repair and re-manufacturing of printed circuit boards requires the removal of the conformal coatings, which typically includes application of a solvent-based stripper and subsequent scraping of the coating [1]. During this process, workers are exposed to hazardous volatile solvents. In fact, numerous occupational deaths caused by acute dichloromethane (DCM) poisoning during coating removal operations have been reported since 1980 [2]. Hence, the health hazards of various toxic solvents have aroused the interest of both regulatory bodies (particularly government agencies in Europe and the U.S.) and companies to reduce the overall chemical risks for human health. This is especially true for large scale applications and removal of conformal coatings. Among all conformal coatings used commercially, acrylic conformal coatings account for the largest market segment, representing approximately 50% of total market share [3]. Therefore, this study focused on acrylic coating removal.

It is important to note the significant safety factors that make DCM dangerous and why they are not desirable in industrial use for the removal of conformal coatings. The removal of conformal coatings often takes place in remote locations where workers must strip the coatings by hand, either with brushes or simply by pouring the solvent onto the coating surface [4]. During this process, it is difficult to avoid contact with DCM as workers are not afforded the safety equipment and safety protocols of an industrial chemical processing plant. In some remote locations, DCM’s high density makes it especially difficult for workers to avoid exposure as it tends to evaporate and accumulate in low laying areas. Inhaling DCM vapors is harmful to the human body as DCM can degrade in the blood forming carbon monoxide [2,5]. In addition, long term exposure to the solvent in small doses can also be harmful even in well-ventilated areas as cancers and central nervous system damage have both been strongly correlated with prolonged DCM exposure [6,7,8].

Because of the known effects of DCM, the removal of DCM from commercial and industrial products is an ongoing effort in the U.S. Recent federal regulations (United States Environmental Protection Agency, U.S. EPA) have restricted the use of DCM in consumer use of paint strippers. However, industrial use of DCM in coating removal products, such as conformal coatings, is not restricted.

Recent research in DCM replacement has been spurred forward by the push from governing bodies to remove DCM from use. Researchers and companies have made progress to reduce the use of DCM in certain applications [9]. For example, 3M applied a water and dimethyl adipate mixture as a paint stripper and varnish remover [10]. In addition, Taygerly et al. (2012) proposed a system containing ethyl acetate, ethanol, and heptane to replace DCM in chromatography applications [11]. Aside from certain solvent products developed for specific applications, a methodology has been explored to satisfy different requirements as well. For instance, combining group contribution methods, Hansen solubility parameters (HSP), and thermodynamics, Jhamb et al. (2020) developed a computer-aided model to design sustainable binary solvent mixtures for residential organic coating formulations [12]. However, discussions about this model were limited by the experimental results available in literature. Moreover, no discussion about designing greener solvents for conformal coatings was covered.

Still, several challenges remain for complete removal of DCM. For many companies, the low cost and broad range of applications offered by DCM as well as its fast dissolution time provide support for its continued use. In addition, evaluating safer solvent alternatives requires companies to dedicate large amounts time and resources into the effort. Given the thousands of potential solvents to evaluate and the numerous types of coatings used in industry, this poses a significant challenge for companies with many research and development priorities. Thus, the focus of this research was to provide a high-throughput evaluation method for the data processing of solute dissolution testing and generation of possible safer alternative solvent formulations to DCM for the removal of conformal coatings.

Solubility parameters are numerical estimates used to characterize the degree of molecular interaction between compounds. The Hildebrand solubility parameters were refined by Charles Hansen into the Hansen solubility parameters (HSP), which can be used to predict the solubility of a targeted polymer in a solvent through their polar, hydrogen bonding, and dispersion parameters [13]. If a solvent point in three dimensional (3D) HSP space is located inside the solubility sphere of a given polymer, it is expected that the polymer will have solubility with the solvent (see Figure 1 as an example). For instance, HSP theory was used to successfully identify solvents to dissolve computer housing material containing three different polymers (polystyrene, polycarbonate, and styrene-acrylonitrile) [14].

The novelty of this study is to present a simple but efficient and tailorable method for formulating safer solvents to suit the need for many different applications using conformal coatings. Using HSP theory to identify safer solvents to remove conformal coatings has not been extensively studied before, though it is acknowledged that HSP theory has been used in scientifically formulating or removing consumer painting. In order to screen the most effective solvent formulations for the removal of conformal coatings, a combination of HSP theory and solubility tests was used in this study. This research included the identification of potential safer solvents or solvent blends and the in-depth evaluation of these materials. The objectives of this research were to: (1) Identify potential alternative solvents, including single solvent or solvent blends, to replace DCM using the Hansen solubility parameters (HSP) approach; (2) screen and sort single solvents or solvent blends in terms of safety, toxicity, and cost-effectiveness and (3) carry out performance testing of the selected solvents. The alternative materials were required to be safer from an environmental, health, and safety standpoint, as well as provide equivalent or better stripping performance than DCM. Accordingly, a set of data processes combined with solvent safety considerations was developed to identify suitable solvents. Several solvent blends and some single solvents were screened out successfully using the screening process. The results of this research will be of value to electronics and coating companies using DCM in various conformal coating stripping operations. By safely removing conformal coating, delicate electronics would be available for re-manufacturing, enabling a circular economy.

## 2. Materials and Methods

### 2.1. Materials

The printed circuit boards were obtained from Chanzon (Shenzhen, China). The acrylic conformal coating (contents: methyl ethyl ketone, ethylene glycol, methyl methacrylate, N-butyl methacrylate) was purchased in liquid form from M.G. Chemicals (Needham, MA, USA). The DCM-based coating stripper, Klean Strip Premium and Klean Strip X, were purchased from a local hardware store in Massachusetts (The Home Depot, Tewksbury, MA, USA). The composition of two coating strippers is listed in Appendix A. The ultraviolet (UV) torch (with a high power beam 385–395 nm, 68 Light-emitting diode, LED, models) was obtained from uvBeast (Sutton, UK). All solvents used for this research including purity are listed in Table 1.

### 2.2. General Procedures for Optimization of Solvent Blends

The Hansen solubility parameters have provided a surface energy-based system for predicting solubility phenomena for decades. Three parameters are included to describe the surface energy of a chemical: dispersion forces, including dipole-dipole interactions, (δD), polar forces (δP), and hydrogen bonding force (δH) (Figure 1) [15]. As a result, polymer solubility can be defined as a sphere constructed in the three-dimensional Hansen space with the measured center and radius.

In general, solvents have absolute HSP values (with specific δP, δD, and δH values), while solutes have a sphere that indicates a spatial region of likely solubility (with a radius, R_0_) [16]. The HSP distance from the center of the sphere to a solvent is defined as R_a_ and can be calculated using Equation (1) below:(1)$$R_{a}^{2}=4*{\left(\delta_{D2}-\delta_{D1}\right)}^{2}+{\left(\delta_{P2}-\delta_{P1}\right)}^{2}+{\left(\delta_{H2}-\delta_{H1}\right)}^{2},$$

In Equation (1), $\delta_{D2}$ and $\delta_{D1}$ represent the dispersion forces for a selected solvent and a given polymer, respectively. Similar denotation is also applied to $\delta_{P}$ (polarity) and $\delta_{H}$ (hydrogen bonding). With the values of *R_a_* and *R*_0_, one can calculate the relative energy difference (RED) as:(2)$$\mathrm{RED}=\frac{The\;HSP\;distance\;of\;a\;solvent\;to\;the\;center\;of\;the\;solute\;sphere}{The\;radius\;of\;the\;solute\;sphere}=\frac{R_{a}}{R_{0}}$$

Generally, the closer a solvent is located to the sphere center, the stronger the solvation power of a solvent. Any solvents laying outside the sphere are assumed to be unable to dissolve the solute at the given pressure and temperature.

In this study, a solubility test on the acrylic conformal coating with 39 solvents was performed to construct a polymer sphere using HSP values taken from literature [12]. The pre-dried coating was adjusted to 1 cm^2^ square flakes with 1–2 mm thickness and average weight of 0.12 g. The flakes were immersed in 10 mL of each individual solvent within glass tubes creating a conformal coating concentration of 0.012 g/mL. A UV torch was used to determine whether the acrylic conformal coating flake was dissolved or not by detecting the fluorescence within the invisible and transparent conformal coating in the solution. After a two-hour dwell time at ambient temperature without agitation, the dissolution status was assessed and scored following the rules below: 0 was used for those undissolved at room temperature even after a 2 h dissolution time (i.e., bad solvents); 1 was used for those dissolved at room temperature with a short dissolution time (<2 h) (i.e., good solvents); 2 was used for those fully dissolved with a long dissolution time (>2 h) (i.e., weak solvents).

The score of each solvent was then used to calculate the best-fit polymer sphere with HSP values and radius using multi-response optimization algorithms as described in literature [16,17,18].

During later experiments, it was found that only a few common pure solvents had close HSP values to that of the measured acrylic coating. Therefore, two-component solvent blends with promising HSPs were identified to broaden the group of potential solvent formulations. The solvent blends were screened by using an optimization method developed by the research team. More details were provided in the paragraph below.

Based on the available HSP values [16], over 5500 potential solvent blends were identified to dissolve the target conformal coating. To accelerate the solvent selection process, it was necessary to significantly narrow down the number of potential solvent blends for further evaluation. Initial screening of the data consisted of the removal of false mixtures and immiscible solvent blends. False mixtures were solvent blends optimized with less than 10% by volume of either ingredient solvent. These solvent blends were removed because the low mixture ratios indicated that the majority of the solvent blend’s dissolution properties were associated with the dominant solvent. Thus, the results using these blends did not represent an innovative solvent blend solution to replace DCM. In addition, immiscible blends were removed. Solvent mixtures with a solvent distance greater than 15 MPa^1/2^ were considered as immiscible [13,19]. These solvent blends were removed because immiscible solvents used in the same blend cannot form a thermodynamically stable mixture. For example, mixtures containing 1,4-dioxalane and propylene carbonate were removed from consideration as they were calculated to have a distance of 17.6 MPa^1/2^ from the center point of the solubility sphere for the acrylic conformal coating.

Further screening of the data consisted of removing solvent blends with a solvent distance greater than 4 MPa^1/2^ from the center point of the solute’s solubility sphere [20]. These blends were screened out as those closest to the center of the sphere were most likely to demonstrate superior dissolution properties [21,22].

Once all undesirable solvent blends were removed from the dataset, a list of all individual solvents included in the remaining mixtures was generated. Solvents that were considered unsafe in comparison with DCM were removed from the list. The safety level of all solvents was determined using data from the Pharos (https://pharosproject.net/) [21], an online chemical hazard database. Health hazards were considered based on their relationship with the proposed application of the solvent blend for conformal coating removal, the likelihood of human exposure, the confidence shown in scientific literature that potential risks exist, and the significance of that risk potential in terms of human health and the environment. As a result, solvent blends containing an unsafe solvent were also removed from the optimization list. This approach was designed to mitigate the risk of inconclusive scientific evidence for or against the safety of a given solvent propagating into our optimization while conservatively allowing the inclusion of known safer solvents. This process simplified the optimization process down to the two factors of dissolution performance and solvent safety when determining the optimum choice for a solvent solution.

Solvent pricing was also considered to promote the development of more economically feasible solvent blends. The authors gathered pricing information and screened out any solvents that were at least three times more expensive than DCM. Pricing information was taken online from Sigma-Aldrich (St. Louis, MO, USA), a solvent supplier. This was meant to provide a relatively consistent comparison of individual solvent prices. Sigma-Aldrich was chosen because of the relatively high availability of pricing data, quantity of various solvent types, solvent quality, and purity on their website because of their position as a major supplier of laboratory grade chemical solvents. However, solvent prices did not account for significant market factors such as bulk discount pricing, price of other additives used in conformal coatings, and supply chain costs. Thus, the solvent blend prices obtained were only compared within the given dataset and could not be used to estimate market cost without further cost analysis.

### 2.3. Dwell Time Test

A dip coating method was used to coat the 4 cm wide × 6 cm long printed circuit boards (Appendix A). The dip coating method was selected as it is widely used in the industrial coating process to manufacture bulk products [22]. A printed circuit board was immersed perpendicularly into liquid conformal coating material and removed after one second immersion time. Then, the printed circuit board was hung to remove excess coating material and left to dry at room temperature for 48 h to ensure the coating fully dried (note: drying at room temperature for 24 h was suggested by the acrylic conformal coating manufacturer). A washer/gasket (the Home Depot, Tewksbury, MA, USA), was pasted on the conformal coating surface and a sheet of parafilm attached to the back of the printed circuit board to avoid solvent leakage during the dwell test. The test area for the solvent and coating interaction was located inside the gasket and had an area of 572 mm^2^. Solvent blends tested were selected to have different distances to the sphere center from one another. The solvents were blended in advance to conducting the dwell time test. The Klean Strip Premium coating stripper with DCM was poured into the gasket and covered by a lab watch glass (Fisher Scientific, Waltham, MA, USA) to prevent solvent evaporation during the dwell time test. The lab watch glass and solvent blend were removed after various dwell times (2, 3, 5, 10, and 20 min). The researchers then scraped the printed circuit board surface coated with the acrylic conformal coating. Once approximately 90% of the conformal coating was removed (confirmed using UV torch), the test was considered completed (Figure 2). The same steps were performed with solvent blends to determine the dwell times required to remove 90% of the conformal coating from the surface of the printed circuit board.

## 3. Results and Discussion

### 3.1. Determination of the HSP Solubility of Acrylic Conformal Coatings

The HSP solubility was determined based on the dissolution results using a two-hour dwell time. A detailed list of solvents used is available in Table 1. To better visualize the HSP solubility on paper, the 3D sphere plot was converted into a 2D ternary diagram where the axis represents the ratio of one parameter and the three HSP values of the material [23]. The dissolution test results were presented in Figure 3 with the circle in the triangular plot representing the solubility range of the acrylic conformal coating.

Compared to simply excluding inefficient solvents out of the HSP sphere, the scoring system based on three grades (dissolve, undissolved, and fully dissolved with a long dissolution time) provided improved accuracy for the polymer material. Weak solvents adjust the sphere center and radius, increasing tolerance of solubility limit and allowing for additional solvents to be located inside of the sphere. The final sphere position was further modified by testing more solvents on the surface until a minimized deviation of the fitting could be achieved [17,18]. It is found that improving the accuracy of the HSP value of the studied solute (polymer) was critical for accelerating the solvent selection process down the road.

### 3.2. Model Verification and Improvement

In order to verify the accuracy of the HSP value, a dwell time test was carried out to understand if the solvent blend identified using the developed HSP sphere would have similar performance to DCM and DCM-based coating stripper. Because of the small size of the printed circuit boards, a dip coating method was utilized for ease of operation. The preliminary dwell time test was performed with the Klean Strip Premium coating stripper, which contains a high concentration of DCM. Because the dwell time test aimed to compare the performance of the screened solvent blends with a commercialized coating stripper, the dwell time of the Klean Strip Premium coating stripper and DCM was selected as a baseline. The Klean Strip Premium coating stripper and DCM were able to satisfy the substrate exposure requirement within a two-minute dwell time. The duplicate dwell test allowed a range of dissolution times for each solvent blend (Table 2). The solvent blends located inside the HSP sphere were all able to remove coatings in around three minutes while the solvent blends with a RED value of 1.01–1.67 required longer times nearly in a double magnitude. Given that the three-minute dwell time was quite close to the baseline and that the dwell time for solvents outside the solubility range increased significantly, all selected solvent blends within the solubility range were considered to have acceptable performance. However, the acrylic conformal coating in this experiment was susceptible to many solvents and the mass of coating on the printed circuit board area was relatively low. Thus, some solvent blends outside the solubility range were tested effective after a long dwell time.

One method to improve the accuracy of the solubility data provided in Table 2 would be to carry out a sensitivity test to understand which HSP parameters may affect the solubility more dominantly. It is well known that within HSP, polarity and hydrogen bonding affect the HSP value more than the London dispersion force which has less effect on the solubility parameters because of its weak intermolecular attractive strength (1–2 kcal) [24]. This suggests that additional tests using solvents or solvent blends with a wide range of polarity and hydrogen bonding would be helpful. Identifying the magnitude of effect for each parameter on overall solubility would improve the accuracy of the HSP sphere.

In addition, other solubility parameter systems such as Kamlet–Taft parameters and Richardt’s polarity could be considered to describe the solubility in future research. For instance, Kamlet–Taft parameters considers hydrogen bonding solvent-pair blends (i.e., donor/acceptor), which was not included in the HSP theory. Cross referencing two solubility parameters will enable a more reliable prediction of safer solvents and possibly broaden available compositions of safer solvents [25].

### 3.3. Solvent Optimization

HSPs are assumed to be additive via the geometric mean rule. This means that two given solvents can be mixed in a certain ratio to form a solvent blend with an HSP value that represents the average of both ingredient solvents. This relationship is assumed to be linear such that the HSP value for a mixture of any number of solvents can be calculated given the mixture ratio and the original HSPs of each constituent solvent. Based on this, multiple solvent blends could be derived from various pure solvents to ensure anticipated solubility using HSP. Figure 4 shows an example of solvent blend identification using this method. By adjusting the volume ratio of diacetone alcohol (dwell time: 12 min) and methyl cyclohexane (cannot dissolve acrylic coating after 30 min) to 46:54, the formulated solvent blend was quite close to the center point of acrylic conformal coating solubility sphere and was found to dissolve conformal coating with a 3 min dwell time (Appendix A). The two ingredients of a solvent blend do not necessarily need to be out of the solubility circle as long as the identified solvent blends provided more desirable solubility, cost, and the environment, health and safety (EHS) characteristics.

A three-factored approach was used to optimize the solvent blend selection. The three factors that were deemed most important to the success of a potential solvent replacement formulation were solvent distance (assumed to represent the solubility phenomena of the polymer accurately), the safety/toxicity aspects of each chemical, and price. A maximum solvent distance of 4.0 MPa^1/2^ was chosen to increase the likelihood that all solvent blends included would be able to dissolve the conformal coating within two hours [20]. The dominant guiding factor for the removal of solvent blends due to their safety level was the immediate danger to workers handling the solvent during coating removal operations. Conformal coatings are used to protect delicate electronics in harsh environments, and the coatings are only removed during maintenance and rework of the printed circuit board. For many systems, this means maintenance is carried out by hand by workers in the field or in repair shops/factories. Worker exposure through inhalation of solvent vapors, contact with skin, and other physical hazards present the critical, immediate risks from solvent usage. Therefore, safety factors related to immediate human health were prioritized. In all possible cases, environmental and long-term toxicological effects were also considered.

The PHAROS project provides a chemical hazard database for rating the safety of each solvent based on the chemical hazard categories listed in Table 3. In general, the PHAROS system characterizes each solvent based on the four levels of the GreenScreen^®^ Benchmark system (www.greenscreenchemicals.org) where Benchmark 1 is considered a chemical of high concern and should be avoided, and Benchmarks 2 through 4 represent solvents with increasing levels of safety. Any solvent with a Benchmark 1 rating was excluded from further consideration as an alternative [26].

For solvents that have not yet been given a GreenScreen^®^ certification benchmark, PHAROS collects and categorizes chemical hazards into five categories: Group I Human (“chronic or life-threatening human health endpoints” that can be induced at low doses), Group II Human (“human health endpoints that can typically be mitigated”), Ecotox (environmental toxicity), Fate (environmental degradation lifetime and pathways), and Physical (physical hazards such as flammability). To accomplish the goal of protecting immediate human health, Groups I and II Human were prioritized. Thus, any solvent showing significant hazard in any of these two categories was removed, regardless of concentration in the mixture. In many cases, the significance of indications given by the PHAROS database were inconclusive due to the fact that some solvents were shown to have significant dangers while others had uncertain toxicity. Convincible results were derived from published literatures and regulations. In these cases, the agency/research group providing the results on PHAROS was critically examined, giving larger and well-established toxicology agencies higher significance than smaller ones. If even more clarity was needed, the solvents were removed to ensure high standards for safety.

To provide further scientific rigor to check this safety system, the solvents that remained after safety removal were checked against the GlaxoSmithKline (GSK) solvent safety guide (Appendix A) [27]. While the GSK safety guide is limited in the included solvents, the available safety scores for the solvents included in the optimization were higher than that of DCM (Health score: 4), which ascertains the health considerations for optimized solvent blends (note: a higher GSK score would be desired when designing safer solvents).

The process of determining solvent safety rating provided an understanding as to which solvent formulations would serve as better replacements for DCM. It provided a framework for an iterative process of solvent screening based on large databases of toxicological information that are available to the public. The protocol for solvent removal, combined with the removal of solvent blends with large distance from the HSP sphere center point can be used to significantly reduce the number of potential solvent blends that require further testing.

The final factor used in the optimization process was price. The price of each solvent was based on a one-liter quantity from Sigma Aldrich during the time of research in 2020. The referenced prices were the lowest prices available for a one-liter quantity while maintaining a minimum of 97% purity. The pricing of solvents can vary considerably, depending on the quantity and purity of the order. This provided a comparable, standard dataset that gave an estimation of how potential solvent blends could compare to each other for development of a marketable solvent formulation. Because the solvent pricing data were not sourced from a bulk solvent market, they cannot be used to directly estimate the price of production of a novel formulation.

An important aspect of the price that was not included in this optimization was the effect of safer chemical usage on companies’ overall costs. When safer chemicals are used, companies can save money on personal protective equipment (PPE), hazardous waste management, insurance costs, and other safety precautions that are not required when using a safer chemical formulation. For this reason, it was not necessary for the price of the safer solvents to be comparable to the cost of DCM. Rather, formulations priced up to 30% higher than DCM were still considered competitive. A short list of solvent blends with both an HSP distance within 2 MPa^1/2^ and having close price to DCM are showed in Table 4. Three solvent blends were tested to verify their solubility performance by the method for dwell time test. Similarly, compared to the two-minute dwell time, all optimized solvent blends in Table 4 were believed to be effective and efficient.

### 3.4. Implications for Similar Conformal Coatings

Acrylic conformal coatings are a relatively economical choice for use in mild environments in which removal processes are simple and fast with various solvents. Other types of conformal coatings are utilized in harsh environments to improve the durability of electronic devices. The different working requirements determine the selection of conformal coatings. Specific conformal coatings with their characteristics are listed in Table 5.

Compared to acrylic coatings, polyurethane and silicon-based coatings are quite difficult to remove. The urethane linkages found in polyurethane coatings and the silicon-oxygen backbone chains found in silicone coatings lead to good moisture and solvent resistance, making them quite difficult to remove as compared to acrylic coatings. Thus, the removal of polyurethane- and silicon-based conformal coatings usually requires specialized solvents, longer dwell times, and an agitation process such as scraping or ultrasonic cleaning. Similar to acrylic coatings, polyurethane and silicone coatings are also one-part coatings (i.e., one resin base diluted with solvents). It is expected that similar procedures used in this study could be applied to the reduce chemical hazards of their strippers as well. However, a longer dissolution dwell time probably would likely be needed for removal of these two coating types. Given the fact water absorption affects the interaction between polymers and solvents considerably, precise control on the drying process prior to testing would also need to be considered. Specific dissolution testing followed by similar solvent optimization methodology can be designed for polyurethane and silicone conformal coatings only after the abovementioned concerns are taken into account. Since conformal coatings may have overlap in their solubility ranges, it is also possible to use the solvent optimization methodology developed from this study to formulate universal solvent blends for removing multiple types of conformal coatings. In this case, the HSP junction point of acrylic, polyurethane, and silicone conformal coatings can be set as the optimization target for identifying universal solvent blends. The universal solvent blends simplify the logistical aspects of the process, making the recovery of the spent solvents feasible.

Epoxy coatings have excellent temperature and chemical resistance, which poses great difficulty for their removal during re-manufacturing. Solvents capable of removing epoxy coatings often also remove the epoxy adhesives in printed circuit boards, causing serious damage to electronic components. In this case, re-manufacturing using a large quantity of solvents would not be appropriate. Therefore, removal of epoxy coatings often involves either thermal methods, grinding and scraping methods, or micro-blasting methods [28].

Solvent recovery and recycling are important aspects of bulk coating removal and electronic re-manufacturing process [29,30]. Although the distillation process is already a well-developed and widespread process for liquid mixtures, the separation of coating stripper and the residual coating material would be difficult due to the close boiling points of coating components and the solvents (e.g., methyl cyclohexane and methyl methacrylate both have the boiling points at about 101 °C.) [9]. As a relatively simple and efficient method, the membrane process has been extensively researched as a method of solvent recovery, driving process intensification. Via the membrane process, the high molecular weight polymeric coatings can be concentrated by nanofiltration, enabling simultaneous solvent recovery and separation of polymeric components via pervaporation [9,31].

## 4. Conclusions

This study demonstrates that the Hansen solubility parameter theory is a straightforward model that can be used to measure the solubility of conformal coatings and determine the compatibility of a solute and solvent. The data processing methods outlined in this study are efficient for the interpretation and manipulation of a long and repetitive solvent optimization list. Although chemical safety considerations were complicated and varied from source to source, the identified alternative solvent blends were found to be comparative to DCM in terms of solubility performance and price without the health risks associated with DCM usage. The results will be of value to improve worker safety during acrylic conformal coating removal processes. It is also expected that using a combination of dissolution testing and solvent optimization methodology can help design safer solvents for polyurethane and silicone conformal coating removers as well. In addition, since conformal coatings may have overlap in their solubility ranges, it is also possible to use the solvent optimization methodology to formulate universal solvent blends for removing various conformal coatings so that the use of multiple complex coating strippers can be avoided. By safely removing conformal coatings, electronic components would be available for re-manufacturing, thus providing progress towards a circular economy.

## Figures and Tables

**Figure 1 polymers-13-00937-f001:**
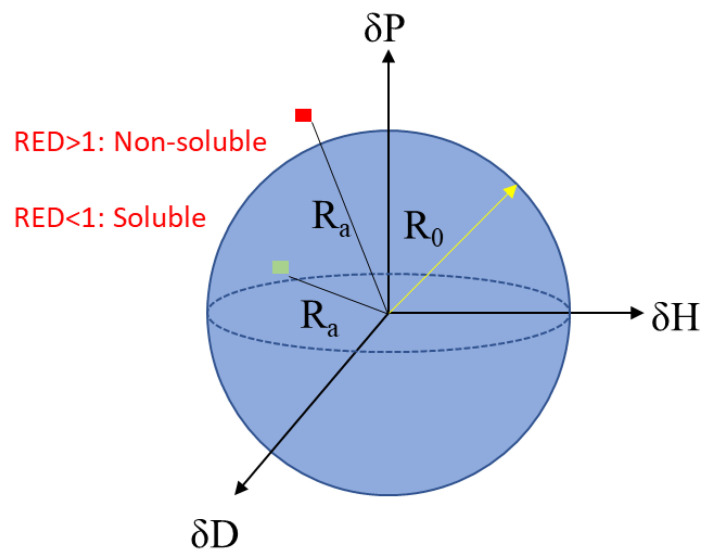
Hansen solubility parameter (HSP) sphere: Red square represents a solvent with a relative energy difference (RED) value > 1 (i.e., theoretically non-soluble solvents); and the green square represents a solvent with a RED value < 1 (i.e., theoretically soluble solvents).

**Figure 2 polymers-13-00937-f002:**
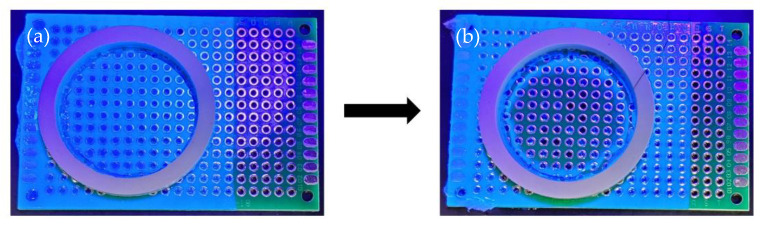
(**a**) Prior to dwell time test with 0% substrate exposure. (**b**) After dwell time test with 90% substrate exposure.

**Figure 3 polymers-13-00937-f003:**
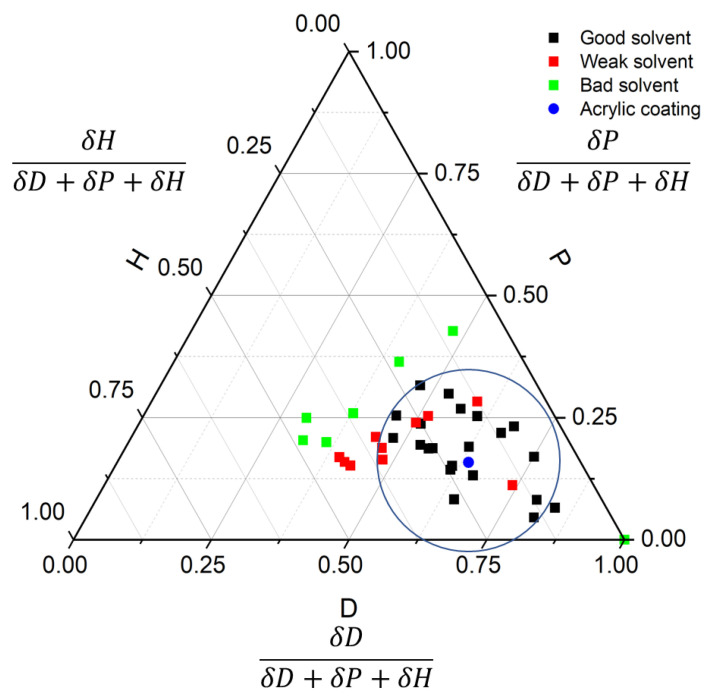
The Hansen solubility parameters (HSP) values (D = 16.5, P = 4.1, H = 5.3, with a radius of 6.8) of the acrylic coating determined by testing 39 solvents.

**Figure 4 polymers-13-00937-f004:**
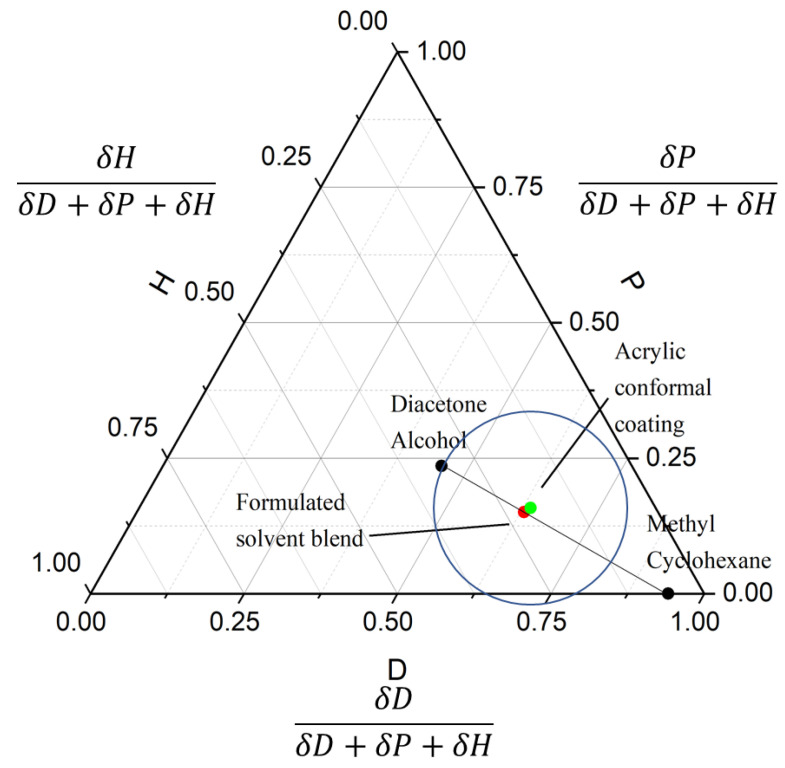
Triangular plot demonstrating the solvent optimization process: By combining diacetone alcohol (a weak solvent with a dwell time of 12 min) and methyl cyclohexane (a bad solvent that cannot dissolve acrylic coating after 30 min), the solvent blend (within the radius) became a good solvent (with a dwell time of 3 min, see more discussion in Appendix A).

**Table 1 polymers-13-00937-t001:** The 39 solvents used in the dissolution test (t = 2 h).

Solvent	Purity	HSP			
		δD	δP	δH	Score ^a^
Toluene	99%	18	1.4	2	1
Dimethyl Carbonate	98%	15.5	8.6	9.7	1
p-Xylene	98%	17.8	1	3.1	1
Benzyl Alcohol	99%	18.4	6.3	13.7	2
Methyl Acetate	99%	15.5	7.2	7.6	1
Ethylene Glycol	99%	17	11	26	0
Undecane	99%	16	0	0	0
Ethyl Acetate	≥99.5%	15.8	5.3	7.2	1
Methanol	≥99.8%	14.7	12.3	22.3	0
Ethanol	≥99.5%	15.8	8.8	19.4	0
1,3-Dioxolane	99%	18.1	6.6	9.3	1
Diethyl Carbonate	99%	15.1	6.3	3.5	1
1-Propanol	99.5%	16	6.8	17.4	2
2-Propanol	99%	15.8	6.1	16.4	2
Propylene Carbonate	99.5%	20	18	4.1	0
Thiophene	99%	18.9	2.4	7.8	1
Propylene Glycol Monomethyl Ether	99.5%	15.6	6.3	11.6	2
Dimethyl Sulfoxide (DMSO)	≥99.7%	18.4	16.4	10.2	0
Acetone	99%	15.5	10.4	7	1
1-Butanol	99%	16	5.7	15.8	2
Dimethyl Glutarate	98%	16.1	7.7	8.3	2
Anisole	99%	17.8	4.4	6.9	1
Ethylene Glycol Butyl Ether Acetate	98%	15.3	7.5	6.8	2
Ethyl Lactate	99%	16	7.6	12.5	2
Diethyl Ether	99%	14.5	2.9	4.6	1
Butyl Benzoate	98%	18.3	5.6	5.5	1
Hexane	99%	14.9	0	0	0
Tetrahydrofuran (THF)	99%	16.8	5.7	8	1
n-Butyl Acetate	99.5%	15.8	3.7	6.3	1
Chlorobenzene	99%	19	4.3	2	1
Diethylene Glycol Monobutyl Ether	97%	16	7	10.6	1
o-Dichlorobenzene	99%	19.2	6.3	3.3	1
Methyl Ethyl Ketone (MEK)	99.5%	16	9	5.1	1
Cyclohexanone	99%	17.8	8.4	5.1	1
Acetophenone	99%	18.8	9	4	2
1-Bromonaphthalene	97%	20.6	3.1	4.1	2
1-Chlorobutane	99%	16.2	5.5	2	1
Formic Acid	99%	14.6	10	14	0
Tetrahydronaphthalene	97%	19.6	2	2.9	1

^a^ Score criteria, 0: undissolved (i.e., bad solvents), 1: dissolved at room temperature within 2 h (i.e., good solvents), 2: fully dissolved with > 2 h (i.e., weak solvents).

**Table 2 polymers-13-00937-t002:** Dwell time of selected solvent blends (n ≥ 2).

Solvent A	Solvent B	Vol. % of A	Vol. % B	δD	δP	δH	Distance(MPa^1/2^)	RED	Dwell Time Needed to Remove > 90% of the Conformal Coating
Butyl Benzoate	Ethyl Acetate	36	64	15.8	5.3	7.2	1.9	0.28	2 min
Anisole	Acetone	83	17	17.4	5.4	6.9	2.8	0.41	2–3 min
Acetone	Butyl Benzoate	14	86	17.9	6.3	5.7	3.6	0.53	3 min
Butyl Benzoate	DMSO	78	22	18.3	8.0	6.5	5.4	0.80	3–5 min
Acetone	1-propanol	87	13	15.6	9.9	7.1	6.3	0.93	3 min
DMSO	Methyl Acetate	33	67	16.5	10.2	8.5	6.9	1.01	3 min
DMSO	Isopentyl Alcohol	12	88	16.1	6.5	12.9	8.1	1.19	8 min
2-butanol	DMSO	81	19	16.3	7.7	13.7	9.2	1.35	6–7 min
DMSO	2- propanol	33	67	16.7	9.5	14.4	10.6	1.55	6–8 min
DMSO	1-propanol	40	60	17	10.6	14.5	11.4	1.67	8 min
DMSO	Ethanol	80	20	17.9	14.9	12.0	13.0	1.91	>20 min
Premium Stripper				16.9	7.5	7.8	4.9	0.72	2 min
Strip X				16.5	7.1	10.1	4.0	0.59	2 min
DCM				17	7.3	7.1	3.8	0.56	2 min

**Table 3 polymers-13-00937-t003:** Hazards considered in this project (modified from the criteria listed in the PHAROS project, https://pharosproject.net/).

Human Health Group I	Human Health Group II and II	Environmental Toxicity and Fate	Physical Hazards
Carcinogenicity,	Acute Toxicity	Acute Aquatic Toxicity	Reactivity
Mutagenicity, and Genotoxicity	Systemic Toxicity and Organ Effects	Chronic Aquatic Toxicity	Flammability
Reproductive Toxicity	Neurotoxicity	Other Ecotoxicity Studies when Available	/
Developmental Toxicity	Skin Sensitization Respiratory Sensitization	Persistence	/
Endocrine Activity	Skin IrritationEye Irritation	Bioaccumulation	/

**Table 4 polymers-13-00937-t004:** Safer solvents optimized from this study for replacing dichloromethane (DCM) in removing conformal coatings.

Solvent A	Solvent B	Ratio A (Vol. %)	Ratio B (Vol.%)	δD	δP	δH	Distance (MPa^1/2^)	RED	GSK (Solvent A)	GSK (Solvent B)
Butyl Diglycol Acetate	Methyl Cyclohexane	69	31	16	2.8	6	1.781	0.262	N/A	8
Butyl Diglycol Acetate	Cyclohexane	70	30	16.2	2.9	5.8	1.448	0.213	N/A	7
Diacetone Alcohol	Methyl Cyclohexane	46	54	15.9	3.8	5.5	1.288	0.189 *	N/A	8
Diethylene Glycol Monobutyl Ether	Methyl Cyclohexane	49	51	16	3.4	5.7	1.308	0.192	7	8
Butyl Diglycol Acetate	p-Cymene	56	44	16.6	3.3	5.6	0.894	0.131	N/A	N/A
Ethyl Acetate	Methyl Cyclohexane	72	28	15.9	3.8	5.5	1.370	0.201	8	8
Dibasic Esters (Dbe)	Methyl Cyclohexane	60	40	16.1	3.9	5.4	0.840	0.124 *	N/A	8
Methyl Cyclohexane	Propylene Glycol Monomethyl Ether	54	46	15.8	2.9	5.9	1.945	0.286 *	8	N/A

* The performance of these three formulations have been verified via dwell time testing. It took 3 min for these formulas to remove acrylic conformal coatings. These three formulations were selected because they represent solvent blends with all distance ranges which are below 2 MPa^1/2^. N/A: not available in GSK solvent selection guide.

**Table 5 polymers-13-00937-t005:** General characteristics for conformal coatings.

Conformal Coating Types	Characteristics
Acrylic	Ease of rework Moisture resistant Fast drying
Polyurethane	Good dielectric properties Moisture resistant Abrasion resistant Good solvent resistance
Silicones	Wide use temperature range Flexible Moisture resistant
Epoxy	Good dielectric properties Abrasion resistant High use temperature Excellent solvent resistance

## Data Availability

The data presented in this study are available on request from the corresponding author.

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
