# Peer review of "Removing Acrylic Conformal Coating with Safer Solvents for Re-Manufacturing Electronics"

_polymers, 2021, doi:10.3390/polym13060937_

Round 1
Reviewer 1 Report
The manuscript by Lu and colleagues discusses a design methodology for using safer solvents in the electronics industry. The work is novel and interesting, it fits the scope of the journal well, and generally speaking, the arguments are well-built. It will be a nice addition to the field. However, there are a few major and minor points to be taken care of.
1) The Results section has many paragraphs on general information, which does not discusses the actual results obtained by the authors but give an introduction. These should be deleted. The manuscript needs to be more concise. As an example, lines 332-343, can be deleted, or shortened to a single sentence; there is no need for a lengthy introduction to PHAROS/GreenScreen.
2) The purity and grade for all solvents and chemicals used in the study should be given under section 2.1. Materials.
3) The introduction is well-written and discusses important context to the work. However, a paragraph should be dedicated to green solvents with examples of their use and emergence.
4) Some general literature on less harmful / green solvents to discuss are DOIs 10.1016/B978-0-12-809270-5.00020-0 and 10.1021/acs.chemrev.7b00571
5) Figure 2 can be simply put in a sentence, the information in the figure does not qualify for an illustration.
6) Figure 3 is redundant and should be deleted. Everything that Figure 3 shows can be seen in Figure 4. Simply add the annotation on Figure 4, and also include a scale bar and/or test area for reference.
7) The authors selected DMSO for several tests. Would other polar aprotic solvents work that are greener?
8) The prices that were taken into account (line 377) are based on bulk order from the supplier? Based on what grade? The prices of solvents significantly change with the scale of the order as well as the purity. Elaborate on this in the text.
9) The potential recovery/recycling of the spent solvent should be discussed as well. The use of safer solvents is a good initiative but the spent solvent still needs to be recovered. Briefly mention the potential of solvent recovery, and give examples of efficient/green methodologies such as membranes and extraction (10.1021/acssuschemeng.9b04245; 10.1016/j.chroma.2015.10.083).
10) The conformal coatings’ composition varies significantly. Consequently, the question arises, how general and transferable are the findings presented in the manuscript. The methodology presented can be easily implemented to other conformal coatings? Elaborate on the general applicability of the methodology, and emphasize the potential impact.
Reviewer 2 Report
Dear Authors, Congratulations for your fine work. What would be the next step of the research for safer solvents to remove acrylic coating? Perspective of the work could included in the Conclusion section.
Round 2
Reviewer 1 Report
The authors have done a thorough revision, made significant changes to the manuscript, which is now ready to be published.